# Exploring Role Behavior in Restaurant by Grey Model and Grey Structural Model

**Joyce-Hsiu-Yu Chen** [1], **Shu-Hua Wu** [1] , **Ping-Min Lin** [2] **and Hsueh-Feng Chang** [2,*]

1    Department of Food & Beverage Management, National Kaohsiung University of Hospitality and Tourism, Kaohsiung 812301, Taiwan; joyce@staff.nkuht.edu.tw (J.-H.-Y.C.); sue@mail.nkuht.edu.tw (S.-H.W.)
2    Department of Tourism Management, National Kaohsiung University of Science and Technology, Kaohsiung 824004, Taiwan; binming495@nkust.edu.tw
*    Correspondence: anniechang@nkust.edu.tw

**Abstract:** Based on GM (0,N) grey model and grey structure model in grey system theory, this study takes Chinese restaurant in tourist hotel as a case to analyze service role behavior. A total of 241 questionnaires were collected to calculate index weighted coefficients, and then 12 experts carried out an investigation to construct clusters. There were 12 dimensions of professional competencies, and a total of 50 indicator factors were analyzed for role behavior in a restaurant. According to the results, there are three role behaviors for service staff in Chinese restaurants: supportive, interactive, and integrative role behaviors. In theory, this reinterprets the meaning of catering service competencies and defines the role types of catering service staff. In practical applications, restaurant managers could apply this result to help service staff to understand their current role, in order to reduce their role pressure and to increase their job satisfaction and performance.

**Keywords:** GM (0,N) model; grey structural model; role behavior; service competency

**MSC:** 68uxx

## 1. Introduction

Competency is a set of behavioral patterns that the incumbent needs to achieve to perform their tasks and functions [1], and which combines experience, responsibility, knowledge, and skills [2]. Professional competency refers to the attitudes, skills, and knowledge required for a certain position [3,4], which signifies the abilities required by the workplace [5]. Restaurant service staff can, not only provide excellent service, but also link restaurants and customers together. Hence, in addition to placing talent resources as a priority, analyzing the professional competency and role behavior of restaurant service staff is a key factor for success.

The people working in restaurants are mostly frontline service staff. Furthermore, due to the rapid development of the market and the diversification of customers, service staff now are dealing with different challenges from in the past. Wu and Hsiung divided service competency into relation-oriented competency and task-oriented competency: the former includes persuasiveness, listening, communication, and attitude; while the latter contains knowledge application, technical knowledge, and appropriate and efficient methods to solve customers' problems [6]. Rainsbury, Hodges, Burchell and Lay, and Hodges and Burchell, all emphasized the establishment of general competency, but ignored the service competency in the service industry [7,8]. Thus, restaurants can enable their employees to pursue a sense of achievement and job performance satisfaction by establishing a code of service competency.

Brouther believed that an employees' job performance could evaluate how well they complete their tasks [9]. Additionally, how employees behave in the real workplace is named role practice or actual role play [10–12]. In service delivery, the first-line personnel

often exercise their rights and obligations according to their role and position. Therefore, this study explores the professional role behaviors of a restaurant's service personnel while serving customers and their actual behaviors when exercising their role responsibilities.

In previous studies, grey relational grade, regardless of the localized or globalized grey relational grade, were based only on a one-dimensional analysis, resulting in them ranking each other. Moreover, it is impossible to perform a hierarchical object-based analysis in this way. Therefore, this study integrated the localized and globalized grey relational grade of grey system theory. The localized grey relational grade value was taken as the vector of the X-axis and the globalized grey relational grade as the vector of Y-axis; this study expanded the analysis of results from one-dimensional to two-dimensional, and into a mathematical model for hierarchical analysis, which is called a grey model (GM). Based on the weight model GM (0,N) and the grey model of grey system theory [13,14], this study constructed a professional competency cluster for service staff in Chinese restaurants, to assist restaurants in summarizing their role behaviors.

Therefore, the purpose of this study was as follows. First, the GM (0,N) model was used to analyze the importance of professional competency factors. Second, the GSM was used to construct a cluster for role behavior.

## 2. Literature Review

### 2.1. Service Competency

Competency signifies an individuals' abilities [1], a general term for individuals' knowledge, skills, and abilities while performing tasks [15]. In other words, it refers to the efficiency and effectiveness due to an individuals' qualities and talents when engaged in a job [16]. Regarding professional competency, it represents the required ability to perform tasks effectively and completely [3], including comprehensive technical knowledge and behaviors acquired through conceptual learning [17]. Professional competency signifies the capabilities required by the employee to complete their professional job [18,19]. However, regardless of the definition of competency or professional competency, it is necessary for individuals to understand and improve their knowledge and skills in order to finish the required tasks.

Service is intangible in the service industry, and the contact between customers and service staff is very important. Hence, service staff need to understand how to enhance their customer-oriented service abilities. Rainsbury et al. defined service competency as the relevant knowledge and abilities required by the front-line service staff of the service industry while performing tasks to meet customers' demands [7]. According to Wu and Hsiung, service competency can be separated into two factors: "task-oriented function", and "relation-oriented function" [6]. Task-oriented competency refers to the professional knowledge and analytical abilities that front-line service staff in the service industry should have when performing tasks to meet customer demands and to achieve a good overall performance. This includes knowledge application and technical skills, which enable service staff to help customers understand products and services and deal with issues to satisfy their demands. Relation-oriented competency refers to the interactive communication and social skills that can help the service industry's front-line service staff meet customer demands while carrying out tasks, including persuasiveness, listening, communication, and attitude, as well as interpersonal relationship competency. In summary, service competency, which can be enhanced through training or development, is the abilities required of service industry personnel to be competent in a certain position.

Recent researches have focused on the establishment of core competency, professional competency, or management competency in various fields, while rarely discussing the service competency of the service industry. The purpose of this study was to investigate the professional competency of Chinese restaurant service staff and to analyze their role behavior cluster, according to their service competency. It hopes to help restaurant service staff provide high-quality service and contribute more to their department.

## 2.2. Role Behavior

During the process of service delivery, the first-line staff have many opportunities to interact with customers, and their relationship is quite close [20]. The first-line staff in the service industry are important to the organization, providing services through contacting customers and forming an impression of the enterprise. The role performance of employees can be evaluated from their rate of compliance of job objectives [9,21]. When specific behaviors required by the organization are positively rated and rewarded, employees incorporate them into their role schema [22]. With the changes in the market and the awakening of consumer awareness, the roles of service staff are becoming more and more diverse. Employees have to play a specific role in the workplace, and their actual behavior is called "role practice" [10,11]. Traditional human resource management analyzes the role behavior of certain positions from the organization's perspective and establishes a code of conduct through job descriptions. However, the role practice mostly takes place during interpersonal interactions, which can be greatly affected by the code of conduct. Moreover, as the professional work and role of service personnel in restaurants may not receive sufficient attention, the form of service is often reduced to administrative work, resulting in poor service quality. Therefore, this study defined role practice as the abilities and capabilities shown by restaurant service staff while exercising their role responsibilities in their position.

Based on the three main work behaviors in an organization: proficiency, adaptive, and proactive behaviors [23], there should be different levels of requirements regarding the competencies that catering service staff should have: for new recruits, senior personnel, and managers. Zhou and Hsu studied work role behavior and found that there are three kinds of Chinese employees: newcomers, seniors, and the manager. Newcomers must perform and learn well; senior staff must perform, learn, and create well; and the manager must perform, learn, create, and lead well, considering the job they took and the role behavior they must perform [24]. Therefore, from an analysis of competency, combined with role behavior, the role of professional workers will be better demonstrated. This is also helpful for organizations to assign tasks and clarify the role-based behaviors for tasks at different work stages. Therefore, it is imperative to clarify the role behaviors of catering service staff with different qualifications, such as newcomers, the seniors, and managers.

Competency is a label signifying the ability to perform or the behavior needed to perform a role [7,25,26]. A member of staff having good role awareness can help them overcome role pressures, balance role conflicts, and increase performance [25]. Previous research in the hospitality and tourism industry has focused on the analysis of service competency [18,19,27], while there are little studies on roles and role behaviors. Therefore, this research, based on the developed service competency and applying grey relational cluster analysis, is important. This study explores service role behavior and not only extends competency theory, but also contributes to the practical application of catering.

## 2.3. Grey System Theory and GM (0,N) Model

Professor Ju-Long Deng proposed the grey system theory in 1982, in order to research a system model, when conditions and information are not clear and incomplete, through relational analysis, constructing a model for prediction and decision-making [28]. The term weight in statistical methodology refers to the distribution frequency of a factor in the system, which is usually used to analyze proportions [29,30]. With the continuous efforts exerted by researchers, the GM (0,N) model has become more complete for application in various fields and takes "uncertainty", "multivariate input", "discrete data", and "incomplete data" into consideration [31]. Due to the ambiguity and uncertainty of role behavior for catering staff, the GM (0,N) of grey system theory was used in this study for data analysis, mainly because it can effectively deal with uncertainty. Hence, based on Chang et al., this study established 12 dimensions and 50 indicators to analyze the importance of influential factors for professional competencies using the GM (0,N) of grey system theory [19].

### 2.4. Grey Structural Model (GSM)

The grey structural model was put forward by Yamaguchi, Li, and Nagai based on grey theory [14]. It provides data for calculation and plotting through grey relational analysis, and finally forms a directed graph. The grey model integrates the localized and globalized grey relational grades of grey system theory. In other words, it allocates each object to be analyzed to a pair from the coordinate system on a 1D plane and hierarchically orders them using the cut set, to convert the 1D plane to a 2D plane. GSM is suitable for research surveys with various uncertain factors. By interpreting discrete series and presenting the weights and ranking, the GSM can transfer those uncertainties into cluster groups. Hence, compared with other methods, the GSM is more objective.

### 2.5. Grey Clustering

Cluster analysis is a method of classification of observed values or samples according to the values observed or the features of the samples, grouping things in the same cluster that possess high homogeneity, and with things in different clusters possessing high heterogeneity [32]. Clustering is crucial in data analysis and data mining applications. Cluster analysis can be used to discover structures in data without providing an explanation/interpretation [33]. Therefore, this study used the GSM to construct the cluster for role behavior.

### 3. Methods

From studies related to hospitality, it was found that competency analyses and applications for hospitality are lacking of methods. Therefore, this study, based on the research results of Chang et al., applied the GM (0,N) model under the grey system theory on competency dimensions, analyzing the importance of 50 indicators according to the 241 questionnaires completed by the service staff in Chinese restaurants of tourist hotels in Taiwan [19]. The survey was conducted from October to November 2015, 470 questionnaires were sent out, 272 were recollected, and 31 invalid questionnaires (with incomplete data) were excluded; in total 241 were effective and the effective rate was 51.277%.

Subsequently, this study applied the GSM to analyze 12 professional competency dimensions, including Serving for Ordering, Serving Meals, Serving Procedures, Serving Skills, Caring for Diners, Handling Customer Complaints, Managing Customer Relations, Expressing and Communicating Abilities, Tableware Knowledge and Maintenance, Dining Environment Maintenance, Serving Beverages, and Promoting Festive Products. There were 12 experts, who have been working for decades in some of the best restaurants all around Taiwan, and who carried out the complete investigation in March 2019 to construct the cluster for role behavior.

### 3.1. GM (0,N) Model

The primary effect of the GM (0,N) model is the "cardinal relation" among *N* variables, which is an analysis of static factors [29]. The formula of GM (0,N) is as follows:

$$az_1^{(1)}(k) = \sum_{j=2}^{N} b_j x_j^{(1)}(k) = b_2 x_2^{(1)}(k) + b_3 x_3^{(1)}(k) + \ldots + b_N x_N^{(1)}(k) \tag{1}$$

In which:

$$z_1^{(1)}(k) = 0.5x_1^{(1)}(k-1) + 0.5x_1^{(1)}(k), k = 2\,,\,3\,,\,4\,,\,\ldots\,,\,n$$

1. Substitute with different values to arrive at

$$
\begin{aligned}
a_1 z_1^{(1)}(2) &= b_2 x_2^{(1)}(2) + \ldots + b_N x_N^{(1)}(2) \\
a_1 z_1^{(1)}(3) &= b_2 x_2^{(1)}(3) + \ldots + b_N x_N^{(1)}(3) \\
&\ldots\ldots\ldots\ldots\ldots\ldots\ldots\ldots\ldots\ldots \\
a_1 z_1^{(1)}(n) &= b_2 x_2^{(1)}(n) + \ldots + b_N x_N^{(1)}(n)
\end{aligned}
\tag{2}
$$

2. Divide both sides of the equation above by $a_1$ and transform it into the matrix form

$$
\begin{bmatrix}
0.5x_1^{(1)}(1) + 0.5x_1^{(1)}(2) \\
0.5x_1^{(1)}(2) + 0.5x_1^{(1)}(3) \\
\vdots \\
0.5x_1^{(1)}(n-1) + 0.5x_1^{(1)}(n)
\end{bmatrix}
=
\begin{bmatrix}
x_2^{(1)}(2) & \ldots & x_N^{(1)}(2) \\
x_2^{(1)}(3) & \ldots & x_N^{(1)}(3) \\
\vdots & \ldots & \vdots \\
x_2^{(1)}(n) & \ldots & x_N^{(1)}(n)
\end{bmatrix}
\begin{bmatrix}
\frac{b_2}{a_1} \\
\frac{b_3}{a_1} \\
\frac{b_4}{a_1} \\
\vdots \\
\frac{b_N}{a_1}
\end{bmatrix}
\tag{3}
$$

Assuming that $\frac{b_j}{a_1} = \hat{b}_m$ and $m = 2, 3, 4, \ldots, N$, then Formula (3) would become

$$
\begin{bmatrix}
0.5x_1^{(1)}(1) + 0.5x_1^{(1)}(2) \\
0.5x_1^{(1)}(2) + 0.5x_1^{(1)}(3) \\
\vdots \\
0.5x_1^{(1)}(n-1) + 0.5x_1^{(1)}(n)
\end{bmatrix}
=
\begin{bmatrix}
x_2^{(1)}(2) & \ldots & x_N^{(1)}(2) \\
x_2^{(1)}(3) & \ldots & x_N^{(1)}(3) \\
\vdots & \ldots & \vdots \\
x_2^{(1)}(n) & \ldots & x_N^{(1)}(n)
\end{bmatrix}
\begin{bmatrix}
\hat{b}_2 \\
\hat{b}_3 \\
\hat{b}_4 \\
\vdots \\
\hat{b}_N
\end{bmatrix}
\tag{4}
$$

Similarly, utilize the matrix solution $\hat{B} = (Y^T Y)^{-1} Y^T X$ method to work out the value of $\hat{b}_m$, where:

$$
X =
\begin{bmatrix}
0.5x_1^{(1)}(1) + 0.5x_1^{(1)}(2) \\
0.5x_1^{(1)}(2) + 0.5x_1^{(1)}(3) \\
\vdots \\
0.5x_1^{(1)}(n-1) + 0.5x_1^{(1)}(n)
\end{bmatrix}, \;
Y =
\begin{bmatrix}
x_2^{(1)}(2) & \ldots & x_N^{(1)}(2) \\
x_2^{(1)}(3) & \ldots & x_N^{(1)}(3) \\
\vdots & \ldots & \vdots \\
x_2^{(1)}(n) & \ldots & x_N^{(1)}(n)
\end{bmatrix}, \;
\hat{B} =
\begin{bmatrix}
\hat{b}_2 \\
\hat{b}_3 \\
\hat{b}_4 \\
\vdots \\
\hat{b}_N
\end{bmatrix}
\tag{5}
$$

The value of $\hat{b}_m$ represents the weighting of the grey relational ordinal against the standard sequence $x_1$.

### 3.2. Grey Structural Modeling (GSM)

Grey structural modeling mainly integrates the grey relational grade from Yamaguchi et al. [14], including localized grey relational grade and globalized grey relational grade, as shown in Formulas (6) and (7).

$$
\Gamma_{0i} = \Gamma(x_0(k), \; x_i(k)) = \frac{\overline{\Delta}_{\max.} - \overline{\Delta}_{0i}}{\Delta_{\max.} - \Delta_{\min.}}, \; \overline{\Delta}_{0i} = \sqrt{\sum_{k=1}^{n} [\Delta_{0i}(k)]^2}
\tag{6}
$$

After working out all the grey relational grade, calculate and compile all the grey relational grades, to arrive at the matrix $m \times m$, which is called the grey relational matrix *R*.

$$
R_{m \times m} =
\begin{bmatrix}
\Gamma_{11} & \Gamma_{12} & \ldots & \Gamma_{1m} \\
\Gamma_{21} & \Gamma_{22} & \ldots & \Gamma_{2m} \\
\vdots & \vdots & \ddots & \Gamma_{11} \\
\Gamma_{m1} & \Gamma_{m2} & \ldots & \Gamma_{mm}
\end{bmatrix}
\tag{7}
$$

The approach to use the eigen-vector method to sequence and work out the weighting is as follows:

1. Establish the matrix $[R]_{m \times m}$ for the target.
2. Work out the eigenvalue $AR = \lambda R$ for matrix $R$.
3. Work out the eigen-vector for matrix $R$ ($P$): Form

$$P^{-1}AP = diag\{\lambda_1, \lambda_2, \lambda_3, \ldots \lambda_n\}$$

4. Use the eigen-vector corresponding to the maximum eigenvalue $\lambda_{\max}$: The value of the corresponding elements within the eigen-vector shall be the weighting (take the absolute value)

The approach used globalized the grey relational grade as the vector for the *X*-axis, and used the localized grey relational grade as the vector for the *Y*-axis; that is, providing plane coordinates to all the analysis targets. Subsequently, use the concept of cut to provide hierarchy to the plane coordinates and allow the one-dimensional analysis expand into two-dimensional space [34].

### 3.3. Proposed Approach

The proposed process includes the following three steps: (1) Calculate the weights through the grey weight theory; (2) Rank various aspects and optimal behavior indicators; (3) Apply the gray structural model method to construct a role behavior cluster; (4) Classify quality attributes.

## 4. Data Analysis

### 4.1. Competency Dimensions and Indicators

The study used 12 dimensions of professional competency and a total of 50 indicators as factors for consideration. These dimensions and indicators were Serving for Ordering with 5 indicators (k1–k5), Serving Meal with 4 indicators (k6–k9), Serving Procedures with 6 indicators (k10–k15), Serving Skills with 8 indicators (k16–k23), Caring for Diners with 4 indicators (k24–k26), Handling Customer Complaints with 4 indicators (k27–k30), Managing Customer Relations with 4 indicators (k31–k34), Expressing and Communicating Abilities with 3 indicators (k35–k37), Tableware Knowledge and Maintenance with 3 indicators (k38–k40), Dining Environment Maintenance with 4 indicators (k41–k44), Serving Beverages with 3 indicators (k45–k47), and Promoting Festive Products with 3 indicators (k48–k50) [19]. The contents for analysis are set out as shown in Table 1.

**Table 1.** The 12 Dimensions and 50 Indicators of Professional Competency.

| Dimension | Code | Indicator |
|---|---|---|
| Serving for Ordering | k1 | I am able to clearly understand the dishes, preparing procedures, and the signature dish and main dishes of the company, and have inquired and memorized the daily menu in advance, to respond to customers' ordering requirements smoothly. |
| | k2 | When taking orders, I am able to inquire about customers' preference and diet requirements in detail and make explanations and recommendations for dishes regarding their requirements. |
| | k3 | I am able to respond properly according to customers' dining purposes and special requirements (i.e., meat/vegetarian, allergy, religions) and communicate fully with kitchen staff to meet customers' expectations. |
| | k4 | I am able to use my knowledge related to alcoholic drinks and make active recommendation to pair with the dishes, to increase the turnover. |
| | k5 | I am able to remember the order of each table of customers (each customer) and repeat the order again to confirm the dishes, in case of repetition. |

**Table 1.** *Cont.*

| Dimension | Code | Indicator |
|---|---|---|
| Serving Meal | k6 | After the dishes are prepared, I am able to check and determine the tableware, dish, temperature, appearance, and portion, to comply with the quality requirements of safety and hygiene. |
| | k7 | Before serving the dishes, I am able to re-check the dish with customers' order, confirm the sequence of the serving table or room in detail, and serve the dishes in an orderly manner. |
| | k8 | When serving dishes, if the dish or serving sequence is required to be adjusted for some reason, I am able to inform the customer in advance to provide quality service. |
| | k9 | When serving dishes, I am able to place the dish to present the food (i.e., place the main dish to face the guests) and provide explanations to customers regarding special food ingredients or eating manners in detail and in due course, to exhibit professional serving, displaying, and explanation services. |
| Serving Procedures | k10 | I am able to set items ready for work (i.e., tableware, side dishes, and condiments), to ensure the smooth process of service, and in turn to provide a seamless service. |
| | k11 | I am able to follow the "Service SOP" to provide dining services, flexibly adjust the service, sequence, and traffic flow, in due course to control the service rate. |
| | k12 | I am able to replace tableware as required, according to the dishes, and carry out proper services (i.e., serving, splitting dishes, removing dishes) in due course, to execute work more efficiently. |
| | k13 | I am able to detect the problems of "Service SOP" and discuss with the relevant directors to propose viable improvements or adjustment plans. |
| | k14 | I am able to properly pack the unfinished dishes or food for customers to ensure the dishes or food will not leak, and remind customers of eating manner and expiring period. |
| | k15 | After finishing work, I am able to use proper preserving and maintaining methods to keep the supplies and items. |
| Serving Skills | k16 | I am able to welcome and seat customers and see customers out in a polite and accurate manner. |
| | k17 | I am able to arrange seats accurately for guests and provide services based on priorities to meet the dining etiquette for catering. |
| | k18 | I am able to move and set up dining equipment (i.e., table, chairs) in a safe manner and provide efficient work (i.e., holding trays, plates, glasses, and usage of fork/spoon for splitting food) to serve and split dishes to benefit the execution of the subsequent work. |
| | k19 | I am able to use utensils (i.e., holding trays, plates, glasses, and usage of fork/spoon for splitting food) in a safe, hygienic, and efficient manner to provide serving and splitting dishes services, displaying professional services. |
| | k20 | I am able to introduce and recommend cold/hot drinks (i.e., juice, tea, and wine) accurately, and provide services in a smooth and safe manner, and exabit expertise in serving beverages. |
| | k21 | I am able to determine the time to remove dishes and clean the table to complete the removal and after service of meals. |
| | k22 | I am able to master the bill payment procedures effectively and complete the report and statement by myself, to maintain the quality of financial work. |
| | k23 | I am able to understand the demands of customers in the function room and utilize professional skills to control the service timing of the procedures, providing a thoughtful dining experience. |
| Caring for Diners | k24 | I am able to observe the preferences and general condition of customers consuming sauces, tea, or beverages, and prepare, mix, and supplement properly in due course. |
| | k25 | For each contact with customers, I am able to grasp the key and continuity of service, take the initiative to care for customers' dining experience, and make a response or remedies in due course to the improve service quality. |
| | k26 | I am able to evaluate customers' dining experiences and relevant information (i.e., flavor and delicacy of dishes) during the dining period at any time to collect customers' opinions and pass customers' opinions to directors. |

Table 1. *Cont.*

| Dimension | Code | Indicator |
|---|---|---|
| Handling Customer Complaints | k27 | Encountering customer complaints, I am able to stand in the customer's shoes to pacify customers' emotions and listen to customers, allowing the customers to feel respected, so as to soothe their anger and put an end to the customer complaints. |
| | k28 | I will deal with customer complaints with the director according to the guidelines for customer complaints and authorizing degree, and provide feedback or reports regarding customer recommendations, to avoid the continuation of customer complaints. |
| | k29 | I am able to determine the severity of the customer complaint case, control the situation, and decide the timing of reporting to the director for handling. |
| | k30 | I am able to propose the case of customer complaints during a meeting and record it in the meeting minutes for future reference, in case of similar conditions in the future. |
| Managing Customer Relations | k31 | I am able to grasp the opportunities to interact with customers (i.e., discovering common ground), establish a trust relation with customers, to make them my customer base. |
| | k32 | I am able to accurately address customers' last name or titles, identify the characteristics and corresponding requirements of the customers (i.e., birthday surprise), to actively provide more customized services. |
| | k33 | I am able to establish a system to collect customers' opinions (i.e., customer opinion form or staff praise card) and make instant response to improve issues recommended by customers. |
| | k34 | I am able to learn regular customers' consumption habits and dining requirements by heart, interact with them in a more familiar manner, and establish and update customers' information, to maintain long-term friendly relations. |
| Expressing and Communicating Abilities | k35 | I am able to accurately adopt the language or manner used by customers to communicate with customers effectively, so as to provide proper services. |
| | k36 | I am able to adopt listening and observing skills (eye contact or facial expression) to initiate conversations with customers and avoid excessive interference, providing fast services. |
| | k37 | I am able to use accurate hand gestures or body language to provide fast and proper information and services. |
| Tableware knowledge and Maintenance | k38 | When tableware is broken, I am able to record clearly the damaged parts and quantity as a reference for preventing damages, as well as the operating requirements. |
| | k39 | I am able to correctly move, operate, or wipe utensils according to the standards of hygiene and safety, to maintain the cleanness of tableware, allowing customers to use them with ease and avoid excessive damages. |
| | k40 | I am able to check the number of tableware regularly, accurately analyze and report the reason for shortages, and propose effective improvement methods, so as to reduce damage rates and loss rates. |
| Dining Environment Maintenance | k41 | I am able to check the surround environment I am assigned to, clean the floor properly, clean the surrounding environment, doors, and windows of the store to provide a dining space with clean windows and great lighting before opening. |
| | k42 | I am able to inspect the equipment of air-conditioners and lights, as well as the tidiness of item layouts on the tables at any time, handle problems immediately when finding issues, and pay attention to customers' experiences and feelings at the same time. |
| | k43 | I am able to clean up the table immediately once customers have finished dining, to provide a clean and comfortable dining environment. |
| | k44 | I am able to regularly arrange sterilization operations with safety measures (i.e., covering items) to prevent the breeding of vector mosquitoes and ensure the safety and hygiene of the dining environment. |
| Serving Beverage | k45 | I am able to accurately analyze and grasp the features and mouthfeel of different wines to provide explicit explanations to customers. |
| | k46 | I am able to adopt proper opening and breathing manners for wines, according to the features of different wines, to provide professional serving of beverages. |
| | k47 | When providing beverage services, I am able to inform customers regarding the tasting sequence and proper tasting manner, so as to improve the fine wine-tasking experience for customers. |

| Dimension | Code | Indicator |
|---|---|---|
| Promoting Festive Products | k48 | I am able to assist efficiently the organization of selling festive products according to customers' demand, to increase the turnover of the organization. |
| | k49 | I am able to integrate the sales quantity and income for all products accurately, and commence the subsequent sales operations efficiently. |
| | k50 | I am able to deliver products to customers according to customers' demands in the most appropriate and convenient manner. |

Source: [19].

### 4.2. GM (0,N) Model Analysis

For exploring the importance of the factors affecting professional competencies in different dimensions, an analysis was performed according to the items related to each dimension. Five indicators related to serving for ordering was utilized. There were 241 questionnaires in total, and the scores of each indicator are shown in Table 2.

**Table 2.** Weights for Dimension of Serving for Ordering.

| No | k1 | k2 | k3 | k4 | k5 | Top Score |
|---|---|---|---|---|---|---|
| 001 | 5 | 5 | 5 | 4 | 5 | 7 |
| 002 | 5 | 6 | 6 | 6 | 6 | 7 |
| 003 | 5 | 6 | 5 | 5 | 7 | 7 |
| . . . | . . . | . . . | . . . | . . . | . . . | . . . |
| 239 | 6 | 7 | 6 | 5 | 7 | 7 |
| 240 | 5 | 5 | 5 | 4 | 6 | 7 |
| 241 | 5 | 5 | 4 | 4 | 5 | 7 |

All items were measured on a 7-point Likert scale, ranging from 1 (strongly disagree) to 7 (totally agree).

Based on Table 2, we used the GM (0,N) model for analysis, and the grey relational ordinal is as follows:

$x_1^{(0)}$ = top score, $x_2^{(0)}$ = k1, $x_3^{(0)}$ = k2, $x_4^{(0)}$ = k3, $x_5^{(0)}$ = k4 and $x_6^{(0)}$ = k5; therefore, it can produce:

$x_1^{(0)}$ = (7, 7, 7, 7, . . . , 7, 7, 7), $x_2^{(0)}$ = (5, 5, 5, 5, . . . , 6, 5, 5), $x_3^{(0)}$ = (5, 6, 6, 5, . . . , 7, 5, 5), $x_4^{(0)}$ = (5, 6, 5, 6, . . . , 6, 5, 4), $x_5^{(0)}$ = (4, 6, 5, 5, . . . , 5, 4, 5), $x_6^{(0)}$ = (5, 6, 7, 6, . . . , 7, 6, 5).

1. Calculate AGO

$x_1^{(1)}$ = (7, 14, 21, 28, . . . , 1672, 1680, 1687), $x_2^{(1)}$ = (5, 10, 15, 20, . . . , 1364, 1369, 1374), $x_3^{(1)}$ = (5, 11, 17, 22, . . . , 1361, 1366, 1371), $x_4^{(1)}$ = (5, 11, 16, 22, . . . , 1373, 1378, 1382), $x_5^{(1)}$ = (4, 10, 15, 20, . . . , 1282, 1286, 1290), $x_6^{(1)}$ = (5, 11, 18, 24, . . . , 1412, 1418, 1423).

2. Calculate the background value

$z_1^{(1)}$ = ( . . . , 10.5, 17.5, 24.5, 31.5, . . . , 1669.5, 1676.5, 1683.5)

3. Put into the equation to calculate the full value of factors

$$
\begin{bmatrix} 10.5 \\ 17.5 \\ 24.5 \\ \vdots \\ 1669.5 \\ 1676.5 \\ 1683.5 \end{bmatrix} = \begin{bmatrix} 10 & 11 & 11 & 10 & 11 \\ 15 & 17 & 16 & 15 & 18 \\ 20 & 22 & 22 & 20 & 24 \\ \vdots & \vdots & \vdots & \vdots & \vdots \\ 1364 & 1361 & 1373 & 1282 & 1412 \\ 1369 & 1366 & 1378 & 1286 & 1418 \\ 1374 & 1371 & 1382 & 1290 & 1423 \end{bmatrix} \begin{bmatrix} \hat{b}_2 \\ \hat{b}_3 \\ \hat{b}_4 \\ \hat{b}_5 \\ \hat{b}_6 \end{bmatrix}
$$

Use

$$\hat{B} = (Y^T Y)^{-1} Y^T X$$

Among which:

$$X = \begin{bmatrix} 10.5 \\ 17.5 \\ 24.5 \\ \vdots \\ 1669.5 \\ 1676.5 \\ 1683.5 \end{bmatrix}, \hat{B} = \begin{bmatrix} 10 & 11 & 11 & 10 & 11 \\ 15 & 17 & 16 & 15 & 18 \\ 20 & 22 & 22 & 20 & 24 \\ \vdots & \vdots & \vdots & \vdots & \vdots \\ 1364 & 1361 & 1373 & 1282 & 1412 \\ 1369 & 1366 & 1378 & 1286 & 1418 \\ 1374 & 1371 & 1382 & 1290 & 1423 \end{bmatrix}, Y = \begin{bmatrix} \hat{b}_2 \\ \hat{b}_3 \\ \hat{b}_4 \\ \hat{b}_5 \\ \hat{b}_6 \end{bmatrix},$$

The values for $\hat{b}_2 \sim \hat{b}_6$ were calculated, which were 0.3781, 1.4979, 0.9777, 0.3485, and 0.6454, respectively. Meanwhile, the computer's toolbox was used for verification.

Analysis of Other Dimensions

By repeating the methods in the previous section, the same calculation was carried out for all dimensions. Through this calculation, weightings were given for all indicators and we compile all dimensions, then took the sequence from the top three that were higher than 0.05, are as shown in Table 3.

**Table 3.** Weights for GM (0,N) Model Calculation.

| Dimension | Indicator | Weights | Dimension | Indicator | Weights | Dimension | Indicator | Weights |
|---|---|---|---|---|---|---|---|---|
| Serving for Ordering | k1 | 0.3781 | Serving Skills | k16 | 0.1215 | Managing Customer Relations | k31 | 2.1809 |
| | k2 | 1.4979 | | k17 | 0.0320 | | k32 | 0.7994 |
| | k3 | 0.9777 | | k18 | 0.8835 | | k33 | 0.1556 |
| | k4 | 0.3485 | | k19 | 0.0540 | | k34 | 0.0011 |
| | k5 | 0.6454 | | k20 | 0.2154 | Expressing and Communicating Abilities | k35 | 0.2420 |
| Serving Meal | k6 | 0.1153 | | k21 | 0.4764 | | k36 | 1.1326 |
| | k7 | 2.4429 | | k22 | 0.5906 | | k37 | 0.1595 |
| | k8 | 0.9457 | Caring for Diners | k23 | 0.8938 | Tableware Knowledge and Maintenance | k38 | 0.0419 |
| | K9 | 0.2214 | | k24 | 0.6871 | | k39 | 0.3894 |
| Serving Procedure | k10 | 1.9283 | | k25 | 0.4710 | | k40 | 1.6056 |
| | k11 | 0.3940 | | k26 | 0.0495 | Dining Environment Maintenance | k41 | 0.7460 |
| | k12 | 0.2936 | Handling Customer Complaints | k27 | 1.2778 | | k42 | 0.9137 |
| | k13 | 0.1449 | | k28 | 1.2146 | | k43 | 0.5031 |
| | k14 | 0.5235 | | k29 | 0.4936 | | k44 | 0.8507 |
| | k15 | 0.5128 | | k30 | 0.8217 | Serving Beverage | k45 | 0.0248 |
| | | | | | | | k46 | 0.5581 |
| | | | | | | | k47 | 1.8717 |
| | | | | | | Promoting Festive Products | k48 | 1.6738 |
| | | | | | | | k49 | 1.1320 |
| | | | | | | | k50 | 1.5408 |

Indicators in grey were the ones taken for calculating each dimension.

### 4.3. Grey Structural Modeling Analysis

#### 4.3.1. Data Analyzed

For generating the cluster groups of the competency dimensions, an investigation was conducted during March 2019, and questionnaires were filled out by 12 experts who had more than 10 years of relevant practical experience in hospitality. A total of 12 questionnaires were sent and all 12 of the questionnaires were valid; the effective rate was 100%. The average was calculated for each dimension. The average of the important points (0–10) from the dimension of Serving for Ordering (D1) relative to the Serving Meal (D2) was 8.8, for the Serving Procedures (D3) was 8.8, for the Serving Skills (D4) was 8.3, for the Caring for Diners (D5) was 8, for the Handling Customer Complaints (D6) was 8.3, for the Managing Customer Relations (D7) was 8.3, for the Expressing and Communicating Abilities (D8) was 9.1, for the Tableware knowledge and Maintenance (D9) was 6.8, for the Dining Environment Maintenance (D10) was 6.8, for the Serving Beverages (D11) was 8.2, and for the Promoting Festive Products (D12) was 7.2, as shown in Table 4.

**Table 4.** The Average Value of the Relative Important Points from Serving for Ordering Compared to the Other 12 Dimensions.

| NO. | Expert | D2 | D3 | D4 | D5 | D6 | D7 | D8 | D9 | D10 | D11 | D12 |
|-----|--------|-----|-----|-----|-----|-----|-----|-----|-----|-----|-----|-----|
| 1 | E1 | 9 | 9 | 9 | 8 | 10 | 8 | 8 | 9 | 10 | 8 | 4 |
| 2 | E2 | 9 | 9 | 8 | 8 | 8 | 9 | 10 | 6 | 7 | 9 | 8 |
| 3 | E3 | 7 | 8 | 8 | 6 | 3 | 9 | 9 | 2 | 2 | 8 | 8 |
| 4 | E4 | 9 | 9 | 9 | 8 | 8 | 9 | 10 | 6 | 7 | 10 | 10 |
| 5 | E5 | 9 | 9 | 9 | 9 | 8 | 8 | 9 | 8 | 8 | 9 | 8 |
| 6 | E6 | 7 | 8 | 7 | 7 | 7 | 6 | 6 | 7 | 5 | 5 | 6 |
| 7 | E7 | 10 | 10 | 8 | 8 | 9 | 9 | 10 | 8 | 8 | 8 | 7 |
| 8 | E8 | 10 | 10 | 10 | 9 | 10 | 10 | 10 | 9 | 9 | 9 | 9 |
| 9 | E9 | 8 | 9 | 8 | 7 | 6 | 10 | 9 | 5 | 1 | 7 | 5 |
| 10 | E10 | 8 | 8 | 9 | 8 | 10 | 9 | 8 | 8 | 9 | 8 | 8 |
| 11 | E11 | 10 | 8 | 8 | 8 | 10 | 7 | 10 | 8 | 8 | 7 | 7 |
| 12 | E12 | 10 | 8 | 6 | 10 | 10 | 6 | 10 | 6 | 7 | 10 | 6 |
| AVE | | 8.8 | 8.8 | 8.3 | 8 | 8.3 | 8.3 | 9.1 | 6.8 | 6.8 | 8.2 | 7.2 |

E1–E12: code for the experts. D2–D12: the important points (0–10) from the dimension of Serving for Ordering relative to the other 12 dimensions.

#### 4.3.2. Calculation and Analysis

The method above was adopted to calculate the average of the important points (0–10) from one dimension relative to the other 12 dimensions, one by one. These were serving meal, serving procedures, serving skills, caring for diners, handling customer complaints, managing customer relations, tableware knowledge and maintenance, serving beverages, and promoting festive products.

An analysis was made based on the average value obtained from the 12 experts according to all the average for 12 dimensions, which was calculated the same as in Table 4; all the average values are shown in Table 5.

**Table 5.** The Average Value of Relative Important Points from each Dimension Relative to the other Dimensions.

| Vector | Dimension | O1 | O2 | O3 | O4 | O5 | O6 | O7 | O8 | O9 | O10 | O11 |
|--------|-----------|----|----|----|----|----|----|----|----|----|-----|-----|
| $x_1$ | Serving for Ordering | 8.8 | 8.8 | 8.3 | 8 | 8.3 | 8.3 | 9.1 | 6.8 | 6.8 | 8.2 | 7.2 |
| $x_2$ | Serving Meal | 9 | 9 | 9 | 7 | 8 | 8 | 9 | 8 | 7 | 8 | 6 |
| $x_3$ | Serving Procedures | 9 | 9 | 9 | 9 | 8 | 8 | 9 | 8 | 7 | 8 | 7 |
| $x_4$ | Serving Skills | 8 | 8 | 8 | 8 | 8 | 8 | 8 | 8 | 8 | 8 | 6 |
| $x_5$ | Caring for Diners | 8 | 8 | 8 | 8 | 9 | 9 | 9 | 7 | 7 | 8 | 8 |
| $x_6$ | Handling Customer Complaints | 8 | 8 | 8 | 8 | 8 | 9 | 9 | 7 | 7 | 7 | 7 |
| $x_7$ | Managing Customer Relations | 8 | 7 | 7 | 8 | 9 | 9 | 9 | 7 | 7 | 7 | 9 |
| $x_8$ | Expressing and Communicating Abilities | 9 | 8 | 8 | 9 | 8 | 9 | 9 | 7 | 7 | 9 | 9 |
| $x_9$ | Tableware Knowledge and Maintenance | 8 | 8 | 8 | 8 | 7 | 7 | 7 | 7 | 7 | 7 | 6 |
| $x_{10}$ | Dining Environment Maintenance | 7 | 8 | 8 | 7 | 7 | 7 | 8 | 8 | 7 | 7 | 7 |
| $x_{11}$ | Serving Beverages | 9 | 9 | 9 | 9 | 7 | 8 | 9 | 8 | 7 | 7 | 8 |
| $x_{12}$ | Promoting Festive Products | 8 | 7 | 7 | 7 | 8 | 8 | 9 | 9 | 7 | 6 | 7 |

O1–O11: Sequence of average value calculated from each dimension to the other dimensions.

The reference value (11) was taken as a standard ordinal from Table 5, as the data for calculating the grey relational grade.

Therefore, $x_0$ = (10, 10, 10, 10, 10, 10, 10, 10, 10, 10, 10)

And $x_1$ to $x_{12}$ shall be the grey relational ordinal, as follows:

$x_1$ = (8.8, 8.8, 8.3, 8, 8.3, 8.3, 9.1, 6.8, 6.8, 8.2, 7.2), $x_2$ = (9, 9, 9, 7, 8, 8, 9, 8, 7, 8, 6),

$x_3$ = (9, 9, 9, 9, 8, 8, 9, 8, 7, 8, 7), $x_4$ = (8, 8, 8, 8, 8, 8, 8, 8, 8, 8, 6),

$x_5$ = (8, 8, 8, 8, 9, 9, 9, 7, 7, 8, 8), $x_6$ = (8, 8, 8, 8, 8, 9, 9, 7, 7, 7, 7),

$x_7$ = (8, 7, 7, 8, 9, 9, 9, 7, 7, 7, 9), $x_8$ = (9, 8, 8, 9, 8, 9, 9, 7, 7, 9, 9),

$x_9$ = (8, 8, 8, 8, 7, 7, 7, 7, 7, 7, 6), $x_{10}$ = (7, 8, 8, 7, 7, 7, 8, 8, 7, 7, 7),

$x_{11}$ = (9, 9, 9, 9, 7, 8, 9, 8, 7, 7, 8), $x_{12}$ = (8, 7, 7, 7, 8, 8, 9, 9, 7, 6, 7).

(1) Calculate the localized grey relational grade and globalized grey relational grade.

Based on the original sequences from $x_1$ to $x_{12}$, the standard sequence is $x_0$, hence,

(2) Calculate the values sequentially

$\Delta_{01}$ = (1.2, 1.2, 1.7, 2.0, 1.7, 1.7, 0.9, 3.2, 3.2, 1.8, 2.8), $\overline{\Delta}$ = 1.0190

$\Delta_{02}$ = (1.0, 1.0, 1.0, 3.0, 2.0, 2.0, 1.0, 2.0, 3.0, 2.0, 4.0), $\overline{\Delta}$ = 2.000

$\Delta_{03}$ = (1.0, 1.0, 1.0, 1.0, 2.0, 2.0, 1.0, 2.0, 3.0, 2.0, 3.0), $\overline{\Delta}$ = 1.7272

$\Delta_{04}$ = (2.0, 2.0, 2.0, 2.0, 2.0, 2.0, 2.0, 2.0, 2.0, 2.0, 4.0), $\overline{\Delta}$ = 2.1818

$\Delta_{05}$ = (2.0, 2.0, 2.0, 2.0, 1.0, 1.0, 1.0, 3.0, 3.0, 2.0, 2.0), $\overline{\Delta}$ = 1.9090

$\Delta_{06}$ = (2.0, 2.0, 2.0, 2.0, 2.0, 1.0, 1.0, 3.0, 3.0, 3.0, 3.0), $\overline{\Delta}$ = 2.1818

$\Delta_{07}$ = (2.0, 3.0, 3.0, 2.0, 1.0, 1.0, 1.0, 3.0, 3.0, 3.0, 1.0), $\overline{\Delta}$ = 2.1818

$\Delta_{08}$ = (1.0, 2.0, 2.0, 1.0, 2.0, 1.0, 1.0, 3.0, 3.0, 1.0, 1.0) $\overline{\Delta}$ = 1.6363

$\Delta_{09}$ = (2.0, 2.0, 2.0, 2.0, 3.0, 3.0, 3.0, 3.0, 3.0, 3.0, 1.0), $\overline{\Delta}$ = 2.4545

$\Delta_{0(10)}$ = (3.0, 2.0, 2.0, 3.0, 3.0, 3.0, 2.0, 2.0, 3.0, 3.0, 3.0), $\overline{\Delta}$ = 2.5454

$\Delta_{0(11)}$ = (1.0, 1.0, 1.0, 1.0, 3.0, 2.0, 1.0, 2.0, 3.0, 3.0, 2.0), $\overline{\Delta}$ = 1.1818

$\Delta_{0(12)}$ = (2.0, 3.0, 3.0, 3.0, 2.0, 2.0, 1.0, 1.0, 3.0, 4.0, 3.0), $\overline{\Delta}$ = 2.4545

Then $\overline{\Delta}_{max.}$ = 2.5454 and $\overline{\Delta}_{min.}$ = 1.0190

(3) Substitute the data mentioned above into Equation (6), the localized grey relational grades are (0.7182, 0.5881, 0.9252, 0.5469, 0.7837, 0.5064, 0.5266, 1, 0, 0.1177, 0.8066, 01874)

(4) In addition, put the value of $x_1$ to $x_{12}$ into the equation, to work out the overall grey relations, and build up the grey relational grade matrix, as shown below.

$$R_{12\times12} = \begin{bmatrix} 1.0000 & 0.5655 & 0.6536 & \cdots & 0.2923 & 0.4739 & 0.1871 \\ 0.5655 & 1.0000 & 0.5528 & \vdots & 0.4439 & 0.3675 & 0.2254 \\ 0.6536 & 0.5528 & 1.0000 & \cdots & 0.2517 & 0.6536 & 0.1515 \\ \vdots & \vdots & \vdots & \ddots & \vdots & \vdots & \vdots \\ 0.2923 & 0.3367 & 0.2517 & \cdots & 1.0000 & 0.2870 & 0.4343 \\ 0.4739 & 0.3675 & 0.6536 & \cdots & 0.2789 & 1.0000 & 0.1754 \\ 0.1871 & 0.2254 & 0.1515 & \cdots & 0.4343 & 0.1754 & 1.0000 \end{bmatrix}$$

The $\lambda_{max}$ = 5.0190, hence, the corresponding eigen-vectors were (0.3638, 0.2897, 0.3292, 0.3139, 0.3344, 0.3629, 0.2136, 0.2345, 0.2264, 0.2478, 0.2849, 0.1972), and the results of the calculation are shown in Table 6.

**Table 6.** The LGRG and GGRG of Competency Dimensions.

| Dimension | LGRG | GGRG |
|---|---|---|
| Serving for Ordering (D1) | 0.7182 | 0.3638 |
| Serving Meal (D2) | 0.5881 | 0.2897 |
| Serving Procedures (D3) | 0.9252 | 0.3292 |
| Serving Skills (D4) | 0.5469 | 0.3139 |
| Caring for Diners (D5) | 0.7837 | 0.3344 |
| Handling Customer Complaints (D6) | 0.5064 | 0.3629 |
| Managing Customer Relations (D7) | 0.5266 | 0.2136 |
| Expressing and Communicating Abilities (D8) | 1.0000 | 0.2345 |
| Tableware knowledge and Maintenance (D9) | 0.0000 | 0.2264 |
| Dining Environment Maintenance (D10) | 0.1177 | 0.2478 |
| Serving Beverages (D11) | 0.8066 | 0.2849 |
| Promoting Festive Products (D12) | 0.1874 | 0.1972 |

Localized Grey Relational Grade (LGRG)/Globalized Grey Relational Grade (GGRG).

### 4.3.3. Drawing the Diagram

The 12 dimensions of professional competencies were distributed into three clusters after calculation with the GSM. This was as follows: cluster 1: Tableware Knowledge and Maintenance, Dining Environment Maintenance, Promoting Festive Products; cluster 2: Serving for Ordering, Serving meal, Serving Skills, Handling Customer Complaints, Managing Customer Relations; and cluster 3: Serving Procedures, Caring for Diners, Expressing and Communicating Abilities, Serving Beverages. The results of the analysis drawn by the computer's toolbox are shown in Figure 1.

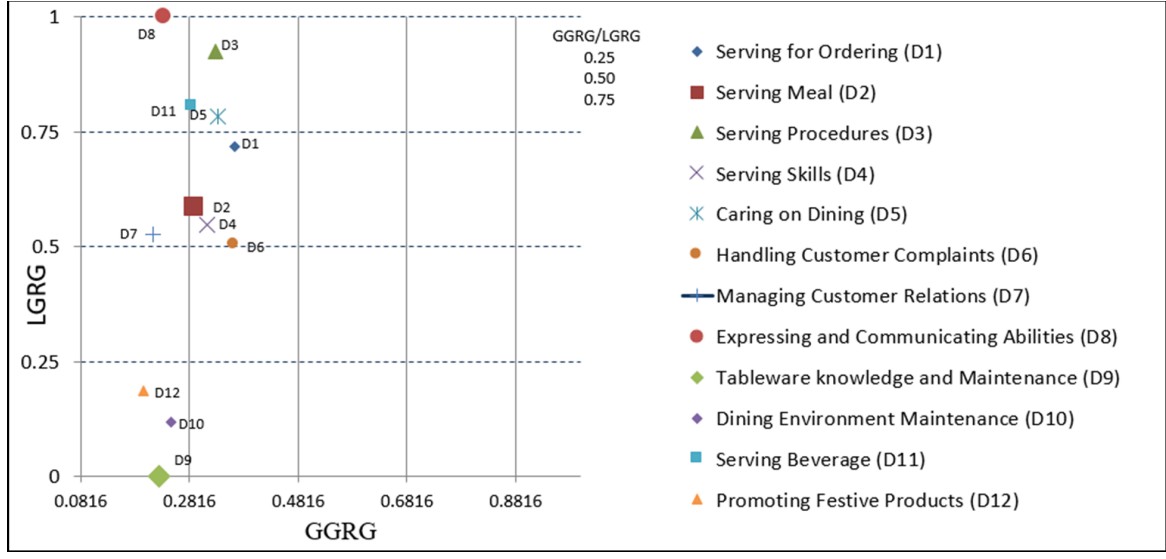

**Figure 1.** Graph of Clusters for Role Behavior.

*4.4. Discussion*

According to the calculation results, the study found that the professional competencies for serving in Chinese restaurants could be divided to three groups. Cluster 1 included three dimensions: tableware knowledge and maintenance, dining environment maintenance, and promoting festive products. These three dimensions were the competencies for side work for staff working at a restaurant. With these kinds of competencies, service staff might not need to connect with the guests directly, but they must have the basic competencies for running a meal time. This cluster of role behaviors in restaurant service was more supportive than presentative. This group was clustered into supporting the service, which were more task-oriented competencies [6]. In addition, this group appeared more likely to be the competency for a newcomer as service staff for restaurants [26]. This group was named supportive role behavior.

Cluster 2 included five dimensions: serving for ordering, serving meal, serving skills, handling customer complaints, and managing customer relations. These dimensions were all related to the guests' dining, which are the main competencies for service staff in meal serving at a restaurant; facing guests and making a good contact while doing service. Cluster 2 was distributed as reacting while doing meal service, which was also a task-oriented relative competency [6]. This group was named interactive role behavior.

This group of competencies is a weapon to help standby service staff face customers, which is required by service staff as an essential part of standard operating procedures. This group of role behaviors in restaurant service was expected when a server displayed good performance in supportive role behavior.

Cluster 3 had four dimensions of professional competencies: serving procedures, caring for diners, expressing and communicating abilities, and serving beverages. The competencies in this group were more than simply practical skills, these were the competencies that should be more proactive; to deal with people, to get along with people, and to use human skills and interpersonal skills. Some of this work might be integrated with all the skills and knowledge needed for meeting the guests' needs. This group was clustered as relating with guests, which were more relation-oriented competencies [6]. Normally, this kind of competency was performed by senior service staff at a restaurant [26]. This group was named integrative role behavior.

The findings for the grouping and clustering of role behavior for service staff at a restaurant are shown in Table 7 and Figure 2.

**Table 7.** Clusters of Role Behavior.

| Cluster | Type | Dimension Distribution |
|---|---|---|
| Supportive Role Behavior | Task-oriented Competency | Tableware Knowledge and Maintenance, Dining Environment Maintenance, Promoting Festive Products |
| Interactive Role Behavior | Task-oriented and Relation-oriented Competency | Serving for Ordering, Serving meal, Serving Skills, Handling Customer Complaints, Managing Customer Relations |
| Integrative Role Behavior | Relation-oriented Competency | Serving Procedures, Caring for Diners, Expressing and Communicating Abilities, Serving Beverages |

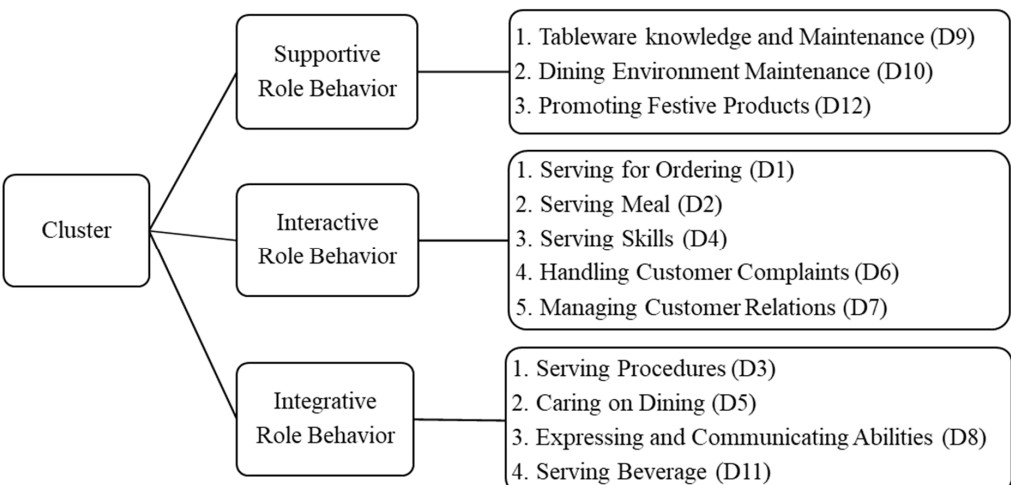

**Figure 2.** Cluster Induction Graph for Role Behavior.

## 5. Conclusions

This paper proposes a new approach for studying service staff's role behavior, based on catering service competencies and using grey system theory. That is unique, as most of previous studies adopted qualitative research methods and rarely adopted quantitative methods for studying the field, and only for competency. The results for the catering service competencies from previous studies were classified by the time sequence of serving a meal [18,19,27]. This study adopted the grey system theory to establish clusters of role behavior that service staff display while performing service work. Through data calculation and analysis, this study determined the weighting coefficients for all indicators and realized the unification of objectivity and subjectivity for weightings; meanwhile, it implemented the GSM analysis method to construct a new model for service role behavior. The study combined the ideal solution with grey relational analysis to reveal the physical significance. Upon applying a comprehensive evaluation of the role behavior of service staff at a restaurant, the new classification revealed the task-oriented role and the relation-oriented role, as found in previous research, which is also in line with the performance of hard skills and soft skills for servers during serving a meal [6]. It is not only clustering the role behaviors for service staff, but also presenting the skills that might help to develop training program and sequences depending on whether they are a beginner or a senior employee.

The new discoveries of the study are as follow:

1. Based on the grey system theory and the weighting calculation, the dimensions and the optimum behavior indicators were ranked. There were 12 dimensions and 50 indicators of catering service competencies for service staff at a restaurant that were confirmed by the grey relational analysis.

2. Through comparisons among GSM, the dimensions were placed into three clusters: supportive, interactive, and integrative role behavior. The first is more task-oriented and is for a newcomer; the second includes both task-oriented and relation-oriented role behavior and is for as a junior or a senior; the last includes more relation-oriented and integrative-oriented role behavior. This finding corresponds to the practical needs of the industry.

3. Detailed explanations and suggestions are as follows:

   (1) Supportive role behavior: this includes the competencies performed in tableware knowledge and maintenance, dining environment maintenance, and promoting festive products. This role behavior encompasses the basic hard skills that a service staff member has in the beginning. In the practical field, before contacting the guest at a restaurant, a server should be trained to understand and maintain the working environment and utensils, called side works,

which guaranty the subsequent service contact and further job training. This is the first and basic role that a new server would be expected to play.

(2) Interactive role behavior: serving for ordering, serving meal, serving skills, handling customer complaints, and managing customer relations. This role is both task-oriented and relation-oriented behavior. The main function of this role for service staff consists in completing the meal serving and guest contact. Therefore, this role behavior would be expected to be well-performed with some soft skills, following the previous role for completing the hard skills. There will be a lot of service contact, with proper reaction to guests' requests, so interpersonal skills would be emphasized when playing this role. As such, when a server is acquainted with the hard skills, then soft skills would be necessary to be learned and performed afterwards, to move from a junior to senior service staff member. Generally, these skills of the role will be trained in the manner of on-the-job training and comply with a dialogue practice. After playing a supportive role, a new server will be trained to perform guest service by serving meals, so as to perform this interactive role behavior well. This is the main role behavior for service staff doing service. A server should be well-trained for the basic hard skills and soft skills to perform this role successfully. Therefore, service staff should learn and understand the menu before providing guests with good service. This includes both of task-oriented and relation-oriented role behavior, which is performed at the scene, as well as with on-the-job training and in the manner of pre-training, complying with the service skills.

(3) Integrative role behavior: serving procedures, caring for diners, expressing and communicating abilities, and serving beverages. This role is completely free from hard skills and is a totally relation-oriented role behavior, to ensure meeting guests' dining experience and sales income. The role for this stage requires integrating the skills and the knowledge for caring and dealing with the needs of guests. The function of this role is mainly played by staff senior or above, such as a supervisor or manager. Therefore, this requires advanced training that mainly targets the shift leaders, supervisors, etc., in the manner of on-the-job training.

4. Regarding the practical contribution to management, on the basis of catering service competency, this study further explores three service staff role behaviors, which transfer the work to the worker itself. This may guide a new method for the industry, regarding practices of work distribution for operation, coordination, and leading, as well as the educational training for new staff and professional development for the existing employees.

The study used GSM to cluster the factors from the data of experts. This method is quite new but seems to function well. In theory, it reinterprets the meaning of the catering service competencies and defines the role types of catering service staff. In practical applications, restaurant managers can apply this result to help service staff understand their current roles, so as to reduce their role pressure and increase their job satisfaction and performance. By strengthening internal service quality, it is possible enhance the value of external customer service. This study only analyzed the catering service competency and role behavior of Chinese restaurants. For further research, a confirmatory analysis could be conducted, and it is suggested that follow-up research could focus on other styles of restaurants, as well as on the tourism and hospitality industry.

**Author Contributions:** Conceptualization, J.-H.-Y.C. and H.-F.C.; methodology, J.-H.-Y.C. and H.-F.C.; validation, J.-H.-Y.C. and H.-F.C.; formal analysis, S.-H.W.; investigation, S.-H.W.; resources, P.-M.L.; data curation, S.-H.W.; writing—original draft preparation, J.-H.-Y.C. and H.-F.C.; writing—review and editing, S.-H.W.; visualization, J.-H.-Y.C. and H.-F.C.; supervision, J.-H.-Y.C.; project administration, H.-F.C. All authors have read and agreed to the published version of the manuscript.

**Funding:** This research received no external funding.

**Institutional Review Board Statement:** Not applicable.

**Informed Consent Statement:** Not applicable.

**Data Availability Statement:** Data are available on request to the authors. The data source is obtained from the questionnaire analysis of the authors' research.

**Conflicts of Interest:** The authors declare no conflict of interest.

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
