# Peer review of "Exploring Role Behavior in Restaurant by Grey Model and Grey Structural Model"

_axioms, doi:10.3390/axioms11070333_

Round 1

Reviewer 1 Report

The paper does not fit journal's format style.

The literature review part is weak and it has to be improved and extended as well.

The following points have to be covered:

-"2.3 Grey System Theory and GM (0, N) weighting model

Professor Ju-Long Deng proposed grey System Theory in 1982 in order to research the system model when the conditions and information are not clear and incomplete through relational analysis," the related work has to be cited.

-In methods part, each presented method has to be followed by an illustrative example, so the reader can use them later.

-The applied model has to be presented clearly step by step.

-The conducted survey has to be presented in the paper.

-Table 1, each indicator has to be cited.

Author Response

Dear Reviewers:

Thanks very much for taking your time to review this manuscript: “Exploring Role Behavior in Restaurant by Grey Model and Grey Structural Model”. We really appreciate all your comments and suggestions. Please find my itemized responses in below (and as attached file as well) and my revisions/corrections in the re-submitted files. 

comments from the reviewer’s

Responses from the Author(s)

The paper does not fit journal's format style.

Thank you for your advice.

We have modified the format of paper to match the format of your Journal.

The literature review part is weak and it has to be improved and extended as well.

Thank you for your advice.
We have already reviewed more reference to highlight the importance of role behavior for catering food service work. (at page 5 by red color).

The following points have to be covered:

-"2.3 Grey System Theory and GM (0,N) weighting model

Professor Ju-Long Deng proposed grey System Theory in 1982 in order to research the system model when the conditions and information are not clear and incomplete through relational analysis," the related work has to be cited.

-In methods part, each presented method has to be followed by an illustrative example, so the reader can use them later.

Thank you for your advice.

We have already cited some references of GM(0,N), please see the manuscript and the reference. (at page 6, 23 by red color)

-The applied model has to be presented clearly step by step.

Thank you for your guidance.

We have added the calculating procedures step by step for LGRG and GGRG. (at page 16-17 by red color).

-The conducted survey has to be presented in the paper.

Thank you for your advice.

We have described the data collecting, such as the questionnaire, etc., in the section of Method. (at page 6-7)

-Table 1, each indicator has to be cited.

Thank you for your advice.

We have cited in Table 1. (at page 12)

Best Regards

Yours Sincerely,

Hsueh-Feng Chang

Reviewer 2 Report

In the current form the paper is written in hard to understand English. Some sentences (for example, the first sentence in the Abstract) are hard to understand at all. Secondly, it is not clear why this paper is important and novel and which really new insight of an important problem it provides. Besides, the methods used in this paper are descibed in a very peculiar, highly technical language, making the paper really hard to understand for greater audience. In my opinion the paper requires significant restructure and rewritting big parts.

Author Response

Dear Reviewers:

Thanks very much for taking your time to review this manuscript: “Exploring Role Behavior in Restaurant by Grey Model and Grey Structural Model”. We really appreciate all your comments and suggestions. Please find my itemized responses in below (and as attached file as well) and my revisions/corrections in the re-submitted files. 

comments from the reviewer’s

Responses from the Author(s)

In the current form the paper is written in hard to understand English. Some sentences (for example, the first sentence in the Abstract) are hard to understand at all.

Thank you for your advice.

The English grammar has been re-edited.

Such as the abstract. (at page 2 by red color)

Secondly, it is not clear why this paper is important and novel and which really new insight of an important problem it provides. Besides, the methods used in this paper are descibed in a very peculiar, highly technical language, making the paper really hard to understand for greater audience. In my opinion the paper requires significant restructure and rewritting big parts.

Thank you for your guidance.

1. We have already reviewed more reference to highlight the new insight of the importance. (at page 5 by red color)

2. We have double checked the paper and restructured the manuscript as the submitted one. (at page 16-17 by red color)

Best Regards

Yours Sincerely,

Hsueh-Feng Chang

Round 2

Reviewer 2 Report

My suggestion have been met at the "acceptable" level, therefore, I can recommend acceptance of the paper.